# RECURRENT REAL-VALUED NEURAL AUTOREGRESSIVE DENSITY ESTIMATOR FOR ONLINE DENSITY ESTIMATION AND CLASSIFICATION OF STREAMING DATA

## ABSTRACT

In contrast with the traditional offline learning, where complete data accessibility is assumed, many modern applications involve processing data in a streaming fashion. This online learning setting raises various challenges, including concept drift, hardware memory constraints, etc. In this paper, we propose the Recurrent Real-valued Neural Autoregressive Density Estimator (RRNADE), a flexible density-based model for online classification and density estimation. RRNADE combines a neural Gaussian mixture density module with a recurrent module. This combination allows RRNADE to exploit possible sequential correlations in the streaming task, which are often ignored in the classical streaming setting where each input is assumed to be independent from the previous ones. We showcase the ability of RRNADE to adapt to concept drifts on synthetic density estimation tasks. We also apply RRNADE to online classification tasks on both real world and synthetic datasets and compare it with multiple density based as well as non-density based online classification methods. In almost all of these tasks, RRNADE outperforms the other methods. Lastly, we conduct an ablation study demonstrating the complementary benefits of the density and the recurrent modules.

## 1 INTRODUCTION

Many tasks in classic supervised machine learning, such as regression and classification, involve processing batched data in an offline fashion: the data, often coming as input-output pairs, is stored first and then used to learn a predictive model for future unseen data. However, many modern applications favor the form where the model update and predict while receiving new data entries. This form is often referred to as learning from data streams. he problem of learning from data streams is closely related to the problem of continual or incremental learning (Losing et al., 2018; Zenke et al., 2017; Lopez-Paz & Ranzato, 2017) which have recently received an increasing interest in the machine learning community

There are three major issues when learning from data streams: memory constrains, concept drifts as well as temporal correlations. The sheer amount of data many modern applications process daily makes it infeasible to store all data and perform offline update of the model (Naeem et al., 2022). In addition, certain data sources do not allow the indefinite hold of the data due to potential privacy regulations (Forti, 2021). Therefore, when learning data streams, it is often assumed that the model only has access to the recent history. Furthermore, concept drifts and temporal correlations are also common challenges when learning from data streams. Under the offline setting, data is often assumed to have the i.i.d. assumption, i.e. each data entry is independently drawn from the identical distribution. However, under the streaming data setting, the independent assumption can be violated, causing temporal correlations in the data, while the violation of the identical assumption can lead to concept drifts problem. These issues often invalidate the model learned from historical data, resulting in further deterioration of its performance.

Density estimation is one of the core tasks in the field of unsupervised learning, branching out to many applications such as classification and clustering. Under the offline setting, Real-valued Neural Autoregressive Density Estimator (RNADE) leverages a neural network parameterized Gaussian mixture model to estimate the density function of real-valued vectors. It is then curious if extending

RNADE to its online form would be possible, namely, the model needs to be updated as new data arrives and we only have a limited amount of history stored in memory. In this paper, we show that the answer is in the positive. Concretely, our contributions are as follows:

1. We propose the Recurrent Real-valued Neural Autoregressive Density Estimator (RRNADE), a versatile density estimator for online learning of data streams.

2. Moreover, we propose an RRNADE based Bayes classifier for online classification of streaming data.

Our model uses a recurrent module to maintain a set of sufficient statistics for the future and capture the potential temporal properties of the data. In addition, it also uses a neural networks parameterized Gaussian mixture model as the density module to compute the conditional density function of the current input given the previous data. We theoretically show that RRNADE is strictly more expressive than Gaussian hidden Markov models Bilmes et al. (1998). We present empirical results demonstrating the ability of RRNADE to adapt to concept drifts and approximating density functions with sequential relations. Moreover, we conduct extensive experiments on various benchmarks of online classification and show that RRNADE outperforms all the compared methods on almost every dataset. In addition, we further demonstrate the importance of both the recurrent module and the density module in the ablation study.

**Related Works**    For online density estimation on streaming data, many of the existing works focus on the adoption of the kernel density estimation (KDE) method (Procopiuc & Procopiuc, 2005; Heinz & Seeger, 2008; Kristan et al., 2011; Boedihardjo et al., 2008). These estimators often relies on maintaining and updating (though merging) a specific number of kernels while incorporating new instances, while in different fashions. In addition to these methods, KDE-Track (Qahtan et al., 2016) leverages an adaptive resampling strategy to deal with concept drifts and improve the estimation accuracy of the KDE-based methods. Another recent method, adaptive local online kernel density estimator (ALoKDE) (Chen et al., 2021), leverages a statistical test for concept drift detection to adapt fast to the concept drift. All these methods can be modified to a classification method via a Bayes classifier.

For online classification on streaming data, there are a number of methods that are direct adaptations of the original offline version to its online case. For example, the online SVM (OSVM) (Li & Yu, 2015), the adaptive random forest (ARF) (Gomes et al., 2017a), can be categorised to this type of methods. In addition, (Liang et al., 2006; Cauwenberghs & Poggio, 2000; Lu et al., 2014) also belong to this class of methods. Other methods like (Bifet & Gavalda, 2007; Bifet et al., 2013) leverage an adaptive window size of the past, (Losing et al., 2016) takes advantage of the short-term and long-term memories, while (Gomes et al., 2017b; Polikar et al., 2001) use ensemble method to further improve the results. Another large class of online classification method is the prototype-based classifiers, such as incremental learning vector quantization (ILVQ) (Losing et al., 2015), generalized LVQ (Sato & Yamada, 1995), robust soft LVQ (Heusinger et al., 2019), and the sparse prototype online kernel density estimator (SPOK) (Coelho & Barreto, 2022).

## 2 BACKGROUND

In this section we will background knowledge including the real-valued neural autoregressive density estimator (RNADE), recurrent models. We will also introduce the formulation of the online density estimation and classification tasks.

**Real-valued neural autoregressive density estimator (RNADE)**    The real-valued neural autoregressive density estimator (RNADE) (Uria et al., 2013) is a generalization of the original neural autoregressive density estimator (NADE) (Uria et al., 2016) to continuous variables. The core idea of RNADE is to estimate the joint density using the chain rule and approximate each conditional density via neural networks, i.e. $p(x_1, \cdots, x_n) = \prod_{i=1}^{n} p(x_i|x_{<i})$ with $p(x_i|x_{<i}) = p_M(x_i|\theta_i)$, where $x_{<i}$ denotes all attributes preceding $x_i \in \mathbb{R}$ in a fixed ordering[*], $p_M$ is a mixture of $m$ Gaussians with parameters $\theta_i = \{\boldsymbol{\beta}_i \in \mathbb{R}^m, \boldsymbol{\mu}_i \in \mathbb{R}^m, \boldsymbol{\sigma}_i \in \mathbb{R}^m\}$. Moreover, we have:

---

[*]Later we will also use the notation $\boldsymbol{x}_{[a,b]}$, where $a < b \in \mathbb{N}$, to denote $\boldsymbol{x}_{a+1}, \cdots, \boldsymbol{x}_b \in \mathbb{R}^d$

$p_M(x_i|\theta_i) = \sum_{j=1}^{m} \boldsymbol{\beta}_i^j \mathcal{N}(x_i|\boldsymbol{\mu}_i^j, \boldsymbol{\sigma}_i^j)$, where $\boldsymbol{\beta}_i^j$ denotes the $j$th element of $\boldsymbol{\beta}_i$, same for $\boldsymbol{\mu}_i^j$ and $\boldsymbol{\sigma}_i^j$ and $\mathcal{N}(x_i|\boldsymbol{\mu}_i^j, \boldsymbol{\sigma}_i^j)$ denotes the Gaussian density with mean $\boldsymbol{\mu}_i^j$ and standard deviation $\boldsymbol{\sigma}_i^j$ evaluated at $x_i$. Note that $\boldsymbol{\beta}_i, \boldsymbol{\mu}_i, \boldsymbol{\sigma}_i$ are functions of $x_{<i}$. These functions are often chosen to be various forms of neural networks. In the classic setting, RNADE with $m$ mixing components and $k$ hidden states has the following update rules:

$$\boldsymbol{h}_i = \boldsymbol{h}_{i-1} + x_i \boldsymbol{w}_i, \quad \boldsymbol{\beta}_i = \text{softmax}(\mathbf{V}_i^\beta \boldsymbol{h}_{i-1} + \boldsymbol{b}_i^\beta) \tag{1}$$

$$\boldsymbol{\mu}_i = \mathbf{V}_i^\mu \boldsymbol{h}_{i-1} + \boldsymbol{b}_i^\mu, \quad \boldsymbol{\sigma}_i = \exp(\mathbf{V}_i^\sigma \boldsymbol{h}_{i-1} + \boldsymbol{b}_i^\sigma), \tag{2}$$

where $\mathbf{V}_i^\beta, \mathbf{V}_i^\mu$ and $\mathbf{V}_i^\sigma$ are $m \times k$ matrices, $\boldsymbol{b}_i^\beta, \boldsymbol{b}_i^\mu$ and $\boldsymbol{b}_i^\sigma$ are vectors of size $m$, and $\boldsymbol{w}$ is a size $n$ vector. The softmax function (Bridle, 1990) ensures the mixing weights $\boldsymbol{\beta}$ are positive and sum to one and the exponential ensures the variances are positive. RNADE is trained to minimize the negative log likelihood: $\mathcal{L}(x_1 \cdots x_n, \theta_i) = -\sum_{i=1}^{n} \log(p_M(x_i|\theta_i))$ via gradient descent.

**Recurrent models** *Recurrent neural networks* (RNN) are a class of neural networks designed to handle sequential data. An RNN takes as input a sequence (of arbitrary length) of elements from an input space $\mathcal{X}$ and outputs an element in the output space $\mathcal{Y}$. In most applications, $\mathcal{X}$ is a vector space, typically $\mathbb{R}^d$. RNNs process sequential data by reading one input at a time and updating a real-valued vector referred to as the *hidden state*. Let $g : \mathbb{R}^k \times \mathbb{R}^d \rightarrow \mathbb{R}^k$ be the transition function between the hidden states at time step $t - 1$: $\boldsymbol{h}_{t-1} \in \mathbb{R}^k$ and at time step $t$. Formally, we have the following definition for RNNs:

**Definition 1.** *Let $\mathcal{X}$ and $\mathcal{Y}$ be the input and output space, respectively. A* recurrent model *with $k$ hidden states is defined by a tuple $R = \langle g, \xi, \boldsymbol{h}_0 \rangle$ where $g : \mathcal{X} \times \mathbb{R}^k \rightarrow \mathbb{R}^k$ is the* recurrent function, *$\xi : \mathbb{R}^k \rightarrow \mathcal{Y}$ is the* output function *and $\boldsymbol{h}_0 \in \mathbb{R}^k$ is the* initial state. *A recurrent model $R$ computes a function $f_R : \mathcal{X}^* \rightarrow \mathcal{Y}$ defined by the (recurrent) relation:*

$$f_R(\boldsymbol{x}_1 \boldsymbol{x}_2 \cdots \boldsymbol{x}_n) = \xi(\boldsymbol{h}_n) \quad \text{where } \boldsymbol{h}_t = g(\boldsymbol{x}_t, \boldsymbol{h}_{t-1}) \text{ for } 1 \leq t \leq n \text{ and } \boldsymbol{x}_1, \boldsymbol{x}_2, \ldots, \boldsymbol{x}_n \in \mathcal{X}.$$

The difference of many kinds of recurrent models often resides in the transition functions. For example, in long short-term memory networks (LSTMs) (Hochreiter & Schmidhuber, 1997), through introducing memory cells and gating mechanisms in the transition functions, it avoids the gradient vanishing/exploding problem that vanilla RNNs often struggle with. In addition, gated recurrent unit networks (GRUs) (Chung et al., 2014) simplify the transition functions of LSTMs and significantly reduce the number of parameters of the model. However, most of these variants focus on the additive relation between the input vector and the hidden state, rarely exploring the multiplicative relation between these vectors. Second order RNNs (2RNN) (Lee et al., 1986) include both the second order (multiplicative) and the first order relations (additive) in the transition functions.

**Online density estimation and classification for streaming data** In this paper, we focus on density estimation and classification for streaming data. Formally, for the density estimation task, let $S = \{\boldsymbol{x}_1, \cdots, \boldsymbol{x}_n, \cdots\}$ be a sequential data stream governed by some distribution $f_t(\cdot)$, where $\boldsymbol{x}_t \sim f_t$ and the subscript denotes the timestamp of the data entry, and $f_t : \mathbb{R}^d \rightarrow \mathbb{R}_0^+$ denotes the distribution at time step $t$. We are then interested in finding an accurate approximation of the density function $f$ at each time step. For a $C$ classes classification task, let $y_t \in \{1, \cdots, C\}$ be the label at time $t$ and $SL = \{(\boldsymbol{x}_1, y_1), \cdots, (\boldsymbol{x}_n, y_n), \cdots\}$ be a sequence of input label pairs. Moreover, the label $y_t$ is drawn from the distribution $q_t$ while $\boldsymbol{x}_t$ is drawn from the distribution $f_t^{y_t}$, i.e. $\boldsymbol{x}_t \sim f_t^{y_t}$ where $f_t^{y_t}(\boldsymbol{x}_t) = p(\boldsymbol{x}_t|y_t)$ denotes the input distribution of class $y_t$ at time step $t$. In this setting, we are interested in predicting the correct label at each time step $t$. As we are approaching these tasks in the streaming setting, it is infeasible to store all the data one have seen so far. Therefore, for both of these tasks, we constrain ourselves to only have access to a short window of data at each time step, e.g. $\boldsymbol{x}_{t-l}, \cdots, \boldsymbol{x}_t$, where $l$ is the window size and controlled to be a relatively small number.

One of the most challenging problems in learning data streams is the concept drift. Intuitively, a concept drift happens when the underlining distribution $f_t$ changes overtime. This change can happen abruptly, when the data changes significantly and occasionally (Iwashita & Papa, 2018). This distribution shift can also occur gradually, when the data values slowly but constantly changes over time. In addition, for classification tasks, concept drifts can occur not only in $f_t^{y_t}$ but also in $q_t$. For example, in video frames classification, the goal is to classify different objects that the current frame contains. In this case, a shift in the camera angle will result in a concept drift in the label's

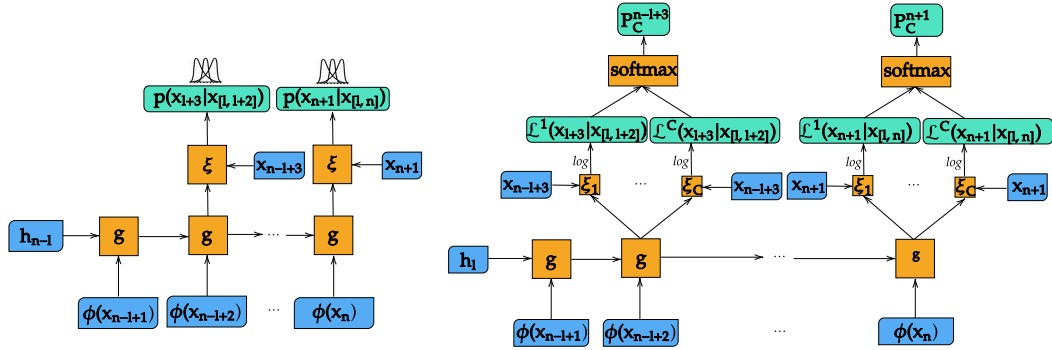

Figure 1: The Recurrent Real-valued Neural Autoregressive Density Estimator for online density estimation (left) and online classification (right), where the window size is $l$ and the likelihood length $p = l - 1$. Blue, orange and green boxes denote input data, functions and outputs, respectively.

distribution, i.e. $q_t$. This shift, naturally can occur abruptly (snap movement of the camera), or gradually (steady movement of the camera), resulting in different types of concept drifts.

## 3 METHODOLOGY

In this section, we introduce the Recurrent Real-valued Neural Autoregressive Density Estimator (RRNADE): a versatile model for density estimation and classification of stream data.

### 3.1 RECURRENT REAL-VALUED NEURAL AUTOREGRESSIVE DENSITY ESTIMATOR (RRNADE) FOR ONLINE DENSITY ESTIMATION

It is natural to wonder if it is possible to use RNADE for *online* density estimation. There are two major issues for this adaptation: 1) the original RNADE model has one set of parameters per feature, which would lead to an infinite amount of parameters to estimate for indefinite length of the stream, and 2) the offline stochastic gradient descent routine needs to be adjusted to its online setting.

To tackle the first problem, instead of approximating the conditionals $p(\boldsymbol{x}_i|\boldsymbol{x}_{<i})$ via the classic RNADE treatment (see Eq. 12), we use a recurrent model $R = \langle g, \xi, \boldsymbol{h}_0 \rangle$ to model the conditionals: $p(\boldsymbol{x}_i|\boldsymbol{x}_{<i}) = f_R(\boldsymbol{x}_1, \cdots, \boldsymbol{x}_i)$. In contrast with RNADE, using a recurrent model allows us to make the state update function independent of the time step, enabling RRNADE to generalize to sequences of arbitrary lengths. Inspired by RNADE, we constrain the output of the recurrent model to be a mixture of Gaussians with diagonal covariance matrices. We now formally introduce the Recurrent Real-valued Neural Autoregressive Density Estimator (RRNADE) model:

**Definition 2.** *A Recurrent Real-valued Neural Autoregressive Density Estimator (RRNADE) with $k$ states and $m$ components is a tuple $\mathcal{R} = \langle g, \phi, \boldsymbol{h}_0, \xi \rangle$, where $\boldsymbol{h}_0 \in \mathbb{R}^k$ is the initial state, $g : \mathbb{R}^k \times \mathbb{R}^{d'} \to \mathbb{R}^k$ is the recurrent module, $\xi : \mathbb{R}^k \times \mathbb{R}^d \to \mathbb{R}_0^+$ is the density module and $\phi : \mathbb{R}^d \to \mathbb{R}^{d'}$ is the input encoder. An RRNADE computes a function $f_{\mathcal{R}} : (\mathbb{R}^d)^* \to \mathbb{R}_0^{+\dagger}$. Given a sequence $\boldsymbol{x}_1, \cdots, \boldsymbol{x}_n$, $f_{\mathcal{R}}$ computes in the following fashion:*

$$\boldsymbol{h}_i = g(\boldsymbol{h}_{i-1}, \phi(\boldsymbol{x}_i)), \qquad \boldsymbol{\beta}_i = \text{softmax}(\mathbf{V}^\beta \boldsymbol{h}_{i-1} + \boldsymbol{b}^\beta) \tag{3}$$

$$\mathbf{M}_i = \mathcal{V}^\mu \bullet_1 \boldsymbol{h}_{i-1} + \mathbf{B}^\mu, \quad \boldsymbol{\Sigma}_i = \exp(\mathcal{V}^\sigma \bullet_1 \boldsymbol{h}_{i-1} + \mathcal{B}^\sigma) \tag{4}$$

$$\xi(\boldsymbol{h}_{i-1}, \boldsymbol{x}_i) = \sum_{j=1}^m \boldsymbol{\beta}_i^j \mathcal{N}(\boldsymbol{x}_i|\mathbf{M}_i^j, \text{diag}(\boldsymbol{\Sigma}_i^j)), \quad f_{\mathcal{R}}(\boldsymbol{x}_1, \cdots, \boldsymbol{x}_n) = \xi(\boldsymbol{h}_{n-1}, \boldsymbol{x}_n) \tag{5}$$

*where $\mathcal{V}^\mu \in \mathbb{R}^{k \times m \times d}, \mathcal{V}^\sigma \in \mathbb{R}^{k \times m \times d}, \mathbf{B}^\mu \in \mathbb{R}^{m \times d}, \mathcal{B}^\sigma \in \mathbb{R}^{m \times d}, \mathbf{V}^\beta \in \mathbb{R}^{m \times k}, \boldsymbol{b}^\beta \in \mathbb{R}^m, \mathbf{M}_i^j = (\mathbf{M}_i)_{j,:} \in \mathbb{R}^d, \boldsymbol{\Sigma}_i^j = (\boldsymbol{\Sigma}_i)_{j,:} \in \mathbb{R}^d, \text{diag}(\boldsymbol{\Sigma}_i^j)$ denotes the diagonal matrix having the components of $\boldsymbol{\Sigma}_i^j$ on the diagonal and $\mathcal{V} \bullet_1 \boldsymbol{h}$ denotes the mode-1 product defined by $(\mathcal{V} \bullet_1 \boldsymbol{h})_{j,k} = \sum_i \mathcal{V}_{i,j,k} \boldsymbol{h}_i$.*

---

[†]$(\mathbb{R}^d)^*$ denotes the set of all possible sequences of arbitrary length constructed with $d$ dimensional real-valued vector at each time step.

As outlined by Equation 5, RRNADE models the conditional density of the current input data, given the history, as a Gaussian mixture. For simplicity, in the rest of the paper, we let $d' = d$ and approximate each conditional via a mixture of Gaussian with a diagonal covariance matrix. This can be changed to a full covariance matrix, should the corresponding assumption (positive semi-definite) of the matrix is satisfied. Note this simplification does not affect the expressiveness of the model, as a Gaussian mixture with a diagonal covariance matrix is also an universal approximator for densities and can approximate a GMM with a full covariance matrix (Benesty et al., 2008), given enough components. Furthermore, although we give specific forms of $\boldsymbol{\beta}_i$, $\mathbf{M}_i$ and $\boldsymbol{\Sigma}_i$ in Definition 2, in practice, one can use any differentiable function of $\boldsymbol{h}_i$ to compute $\boldsymbol{\beta}_i$, $\mathbf{M}_i$ and $\boldsymbol{\Sigma}_i$, so long as $\boldsymbol{\beta}_i$ is positive and sums to one and $\boldsymbol{\Sigma}_i$ is positive definitive. For ease of notation, we will denote by $\mathcal{R}(\boldsymbol{h}, \xi')$ an RRNADE with $\boldsymbol{h}_0 = \boldsymbol{h}$ and $\xi = \xi'$, when necessary.

We show in the following theorem that RRNADE is strictly more expressive than Gaussian HMMs, which are well known for sequential modeling (Bilmes et al., 1998).

**Theorem 3.1.** *Given a Gaussian HMM $\eta$ with $k$ states, there exists a $k$ states $k$ mixtures RRNADE $\mathcal{R}$ with full covariance matrices such that the density function over all possible trajectories sampled by $\eta$ can be computed by $\mathcal{R}$: $p^\eta(\boldsymbol{x}_1 \cdots \boldsymbol{x}_n) = \prod_{i=1}^n f_{\mathcal{R}}(\boldsymbol{x}_{\leq n})$ for any trajectory $\boldsymbol{x}_1 \cdots \boldsymbol{x}_n$. Moreover, there exists an RRNADE $\mathcal{R}'$ such that no Gaussian HMM can compute its density.*

For the second problem, by design, RRNADE approximates a conditional density function of the current input given the history. In the offline setting, one can learn such RRNADE via gradient ascent of the sequences likelihood, i.e, maximizing the likelihood function: $L_{\text{offline}}(\boldsymbol{x}_1, \cdots, \boldsymbol{x}_n) = p(\boldsymbol{x}_1, \cdots, \boldsymbol{x}_n) = \Pi_{i=1}^n f_{\mathcal{R}}(\boldsymbol{x}_{\leq i})$. In the online setting, we assume that we have access to the past data entries from a window of size $l$, i.e., $\boldsymbol{x}_{t-l}, \cdots, \boldsymbol{x}_t$, at each time step $t$. One solution would be to maximize the likelihood over the entire window, i.e. $L(\boldsymbol{x}_{t-l+1}, \cdots, \boldsymbol{x}_t) = \Pi_{i=t-l+1}^t f_{\mathcal{R}}(\boldsymbol{x}_{[t-l,i]})$.

However, this procedure falls short when concept drift occurs. For example, if the data has abrupt drifts (infrequent), then choosing $l$ could be a dilemma: to capture and adapt to this concept drift fast, $l$ is preferable to be small to provide a significant gradient update, which, however, prevents the model from learning temporal dependencies in the data. To address this issue, we propose to have a second hyperparameter $p \leq l$, referred to as the likelihood length, to control the number of past observations that are taken into account in the loss function. That is, RRNADE takes all $l$ past entries as input but the parameters are trained by maximizing the likelihood of only the past $p$ observations. Therefore, not only can we capture temporal dependencies up to the window size $l$, but the choice of $p$ enables us to adjust how fast RRNADE adapt to potential concept drifts in the data. Formally, we maximize the following likelihood function:

$$L_{\text{online}}(\boldsymbol{x}_{t-l+1}, \cdots, \boldsymbol{x}_t) = \Pi_{i=t-p+1}^t f_{\mathcal{R}}(\boldsymbol{x}_{[t-l,i]}) \tag{6}$$

The core idea of this procedure is to obtain and maintain a set of sufficient statistics for the future. At time $t = l$, the model starts with the hidden state $\boldsymbol{h}_0$. After updating the parameters by maximizing the likelihood $\Pi_{i=l-p+1}^l f_{\mathcal{R}(\boldsymbol{h}_0)}(\boldsymbol{x}_{[1,i]})$, the internal state of RRNADE is then updated to $\boldsymbol{h}_1 = g(\boldsymbol{h}_0, \boldsymbol{x}_1)$. At the next time step, $t = l+1$, the first observation $\boldsymbol{x}_1$ is discarded but $\boldsymbol{h}_1$ still represents sufficient statistics of *all* past observations, including $\boldsymbol{x}_1$. The parameters are then optimized to maximize the likelihood $\Pi_{i=l-p+2}^{l+1} f_{\mathcal{R}(\boldsymbol{h}_1)}(\boldsymbol{x}_{[2,i]})$. This process then iterates over the future time steps, carrying the sufficient statistics forward (i.e., at a future time step $t$, even though only the past $l$ observations are stored in memory, $\boldsymbol{h}_{t-l}$ captures sufficient statistics of all past observations).

The training procedure is detailed in Algorithm 1 and a graphical illustration is presented on the left side of Figure 1. We use $L$ and $\mathcal{L}$ to denote the likelihood function and the log likelihood function, respectively. In our experiments, we select $l$ and $p$ using validation over the first $n$ data points. For practical online learning, this can be done by training several RRNADE models in parallel and select the one with the best overall performance after training and predicting on the validation sequence.

## 3.2 RRNADE FOR ONLINE CLASSIFICATION

One straightforward application of approximating densities is online classification. Recall that RRNADE approximates the conditional density of the current input given the history, i.e., $f_{\mathcal{R}}(\boldsymbol{x}_{[t-l,t]}) \simeq p(\boldsymbol{x}_t | \boldsymbol{x}_{<t})$. For the online classification problem, we are interested in the conditional probability of the current label given the history, i.e. $p(y_t | \boldsymbol{x}_1 \cdots \boldsymbol{x}_t)$. Using Bayes rule:

$$p(y_t | \boldsymbol{x}_1 \cdots \boldsymbol{x}_t) \propto p(y_t) p(\boldsymbol{x}_t | \boldsymbol{x}_{<t}, y_t) \simeq p(y_t) f_{\mathcal{R}}(\boldsymbol{x}_{[t-l,t]})$$

---

**Algorithm 1** RRNADE for Online Density Estimation and Classification

---

1: **INPUT**: Input data stream $S = \{\boldsymbol{x}_1, \cdots, \boldsymbol{x}_n, \cdots\}$ for density estimation, or $SL = \{(\boldsymbol{x}_1, y_1), \cdots, (\boldsymbol{x}_n, y_n), \cdots\}$ for $C$ classes classification; window size $l$; likelihood length $p$; a randomly initialized RRNADE: $\mathcal{R}(\boldsymbol{h}_0, \xi_1)$; for classification, extra $C-1$ density modules $\xi_2, \cdots, \xi_C$.

2: **for** $t = l, l+1, \cdots, n, \cdots$ **do**

3:      Compute the log likelihood for each input (and for each class):

4:
$$\mathcal{L}_j^c(\boldsymbol{x}_j|\boldsymbol{x}_{[t-l,j-1]}) = \log(f_{\mathcal{R}(\boldsymbol{h}_{t-l}, \xi_c)}(\boldsymbol{x}_{[t-l,j]})) \text{ for } j = t-p+1, \cdots, t$$

5:      **if** running density estimation task **then**

6:          Compute the sum of the log likelihood over the past $p$ time steps:

$$\mathcal{L}_{online}(\boldsymbol{x}_{t-l+1}, \cdots, \boldsymbol{x}_t) = \sum_{i=t-p+1}^{t} \mathcal{L}_i^1(\boldsymbol{x}_i|\boldsymbol{x}_{[t-l,i-1]})$$

7:          Perform gradient ascent update to $\mathcal{R}(\boldsymbol{h}_{t-l}, \xi_1)$, w.r.t. $\mathcal{L}_{online}$.

8:          Obtain the conditional density estimation at time step $t$:

$$p(\boldsymbol{x}_t|\boldsymbol{x}_{<t}) \simeq f_{\mathcal{R}(\boldsymbol{h}_{t-l})}(\boldsymbol{x}_{t-l+1}, \cdots, \boldsymbol{x}_t)$$

9:      **else if** running classification task **then**

10:         Compute the predicted class distribution: $P_C^j(y_t = c) = \frac{\exp(\mathcal{L}_j^c)}{\sum_{i=1}^{C} \exp(\mathcal{L}_j^i)}$

11:         Obtain the predicted label at current time step: $\hat{y}_t = \arg\max_c P_C^t(y_t = c)$

12:         Perform gradient descent update to $\mathcal{R}(\boldsymbol{h}_{t-l}, \xi_1), \xi_2, \cdots, \xi_C$ w.r.t. the categorical cross entropy: $\frac{1}{p} \sum_{j=t-p+1}^{t} \text{CCE}(P_C^j, y_t)$

13:      **end if**

14:      Update $\boldsymbol{h}_{t-l}$ to $\boldsymbol{h}_{t-l+1}$ via the transition function $g$ of $\mathcal{R}(\boldsymbol{h}_{t-l})$:

$$\boldsymbol{h}_{t-l+1} = g(\boldsymbol{h}_{t-l}, \boldsymbol{x}_{t-l+1})$$

15: **end for**

---

One approach would be to train $C$ different RRNADE models, one for each class: $f_{\mathcal{R}^c}(\boldsymbol{x}_{[t-l,t]}) \simeq p(\boldsymbol{x}_t|\boldsymbol{x}_{<t}, y_t = c)$, where $\mathcal{R}^c$ denotes the RRNADE model for class $c$. However, as we are learning online, each gradient update is often of high variance. This approach introduces too many model parameters, which will further increase the model's variance, resulting in a suboptimal performance. To reduce the number of parameters, we propose to share the recurrent module of all RRNADE models, while keeping the density module specific to each class. RRNADE's prediction at time $t$ is thus given by

$$\hat{y}_t = \arg\max_c [p(y_t = c) f_{\mathcal{R}^c}(\boldsymbol{x}_{[t-l,t]})] \tag{7}$$

For the choice of the prior distribution $p(y_t)$, we recommend using a uniform distribution as extra effort needs to be taken to mitigate the shift in the true prior distribution caused by concept drifts in the data. We defer the study of estimating proper prior for data streams with concept drifts to the future work. We present RRNADE for online classification in Algorithm 1 and a graphical illustration of the model is presented on the right side of Figure 1.

## 4 EXPERIMENTS

In this section, we present empirical results on RRNADE for online density estimation and classification. For density estimation, we experiment with synthetic data to evaluate RRNADE's ability of adapting to concept drifts and to verify Theorem 3.1. For classification, we conduct experiments on both synthetic as well as real world streaming data and compare with multiple density based and non-density based online classification methods. Finally, we show an ablation study to further showcase the significance of both the density module and the recurrent module of the RRNADE. We experimented with three different variants of the RRNADE model. By using LSTM, GRU and

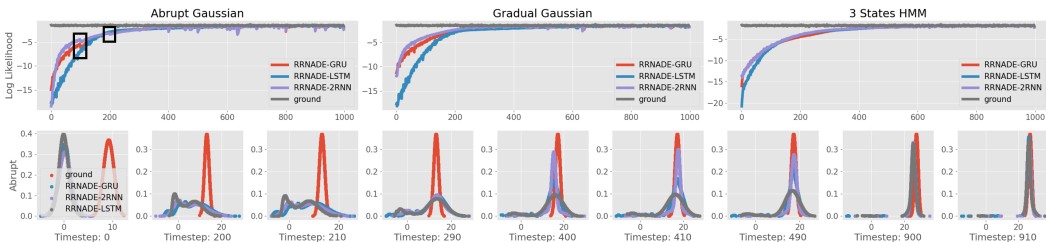

Figure 2: **Top row** (left to right): Log likelihood of (1a) Gaussian with abrupt drift. (1b) Gaussian with gradual drift. (1c) Gaussian HMM. **Bottom row**: Snapshots of learned density at corresponding time steps of Gaussian with abrupt drift, red curve represent the ground truth density function.

| AUC | ALoKDE | oKDE | odKDE | LAIM | KDE-Track | RRNADE-2RNN | RRNADE-LSTM | RRNADE-GRU |
|---|---|---|---|---|---|---|---|---|
| Sea | 83.11 | 78.74 | 75.13 | 77.17 | 81.29 | 88.12 (1.9) | 87.42 (3.6) | **88.88 (2.3)** |
| Hyperplane | 83.44 | 82.36 | 82.36 | 75.56 | 82.8 | 93.12 (2.9) | 93.14 (4.2) | **94.32 (3.1)** |
| Mixed drift | 90.12 | 88.16 | 70.43 | 55.65 | 88.48 | **98.13 (1.2)** | 97.66 (1.1) | 96.11 (1.9) |
| Transient chessboard | 79.6 | 77.4 | 77.08 | 77.18 | 77.98 | 89.16 (3.6) | 86.39 (2.8) | **91.34 (3.2)** |
| Weather | 76.81 | 68.43 | 66.26 | 73.47 | 76.1 | 85.23 (6.4) | **86.67 (3.3)** | 84.14 (2.2) |
| Electricity | 52.51 | 51.57 | 42.45 | 43.25 | 44.06 | **92.14 (4.5)** | 90.68 (5.3) | 91.01 (6.7) |
| Cover type | 97.31 | 57.86 | 96.67 | 93.04 | 96.04 | **98.01 (1.1)** | 96.13 (1.8) | 90.79 (2.7) |
| Poker hand | 91.01 | 88.36 | 82.92 | 82.19 | 91.01 | 94.39 (2.8) | **96.24 (3.4)** | 93.15 (2.9) |
| Rialto | 92.67 | 70.55 | 87.71 | 83.49 | 82.37 | **99.11 (0.9)** | 98.13 (1.5) | 94.35 (0.8) |

Table 1: Averaged online AUC score of RRNADE over 5 seeds (standard deviation in the brackets), compared with (Chen et al., 2021)

2RNN for the recurrent module, we have RRNADE-LSTM, RRNADE-GRU and RRNADE-2RNN, respectively. In all experiments, we use Adam optimizer (Kingma & Ba, 2015).

## 4.1 DENSITY ESTIMATION

To evaluate the performance on density estimation, we first conduct experiments on learning drifting Gaussian to examine RRNADE's ability of adapting to concept drifts. We generate samples from a shifting Gaussian with random initial mean and variance 1. Recall there are two major types of concept drifts, abrupt and gradual drift. For the abrupt drift, the mean is increased by 2 every 100 time steps, while for the gradual drift, the mean is increased by 0.01 every time step. For the model hyperparameters of RRNADE, we set the number of components to be 10, both the window size and the likelihood size is set to 1, number of hidden states of the recurrent module is set to 5 and we use a one layer fully connected neural networks with 5 neurons to be the input encoder $\phi$.

The results are displayed in Figure 2. In Figure 2 (1a) and (1b), we show the log likelihood of Gaussian density function with abrupt and gradual drift on its mean. From the figures, we can see that all three different variants of RRNADE are able to learn the density function and adapt to both of these drifts. The black boxes in (1a) indicates the first two abrupt drifts (time step 100 and 200) of the Gaussian distribution, where visible drops of model performance are observed. Moreover, the adaptation speed increases w.r.t. the time step. The bottom row of Figure 2 shows the learned Gaussain at various time steps. We observe that, at time steps 210 the mean of the mixture model has not been correctly adjusted after 10 time steps of adaptation, while at 410, the model has already adapted to the drift occurred at time step 400.

To verify Theorem 3.1 and show RRNADE can approximate density functions of data streams with sequential features. We generate 1,000 examples from a random Gaussian HMM of 3 states. In this experiment, we set the hyperparameters to be the same as in the above experiment except $l = p = 5$. In Figure 2 (1c), we show the learning curves w.r.t. log likelihood on all three variants. Here we can see all three variants are able to approximate the density function that the HMM emits at each time step. Note there is also a gradual concept drift with the HMM data, as the density function at each time is a mixture of a set of Gaussians, where the mixing weights are the state distributions that the HMM maintains.

| Accuracy | ISVM | LASVM | ORF | ILVQ | LPP | IELM | SGD | NB | RRNADE -2RNN | RRNADE -LSTM | RRNADE -GRU |
|---|---|---|---|---|---|---|---|---|---|---|---|
| Electricity | N/A | N/A | 0.699 | 0.725 | 0.675 | 0.548 | 0.846 | 0.632 | 0.864 (0.02) | **0.878 (0.01)** | 0.858 (0.01) |
| Inter RBF | N/A | N/A | 0.459 | 0.768 | 0.294 | 0.295 | 0.443 | 0.299 | 0.901 (0.02) | 0.922 (0.02) | **0.924 (0.03)** |
| Moving RBF | N/A | N/A | 0.456 | 0.766 | 0.18 | 0.159 | 0.406 | 0.172 | 0.744 (0.01) | 0.739 (0.02) | **0.772 (0.01)** |
| Cover Type | N/A | N/A | 0.896 | 0.883 | 0.397 | 0.513 | 0.946 | 0.546 | **0.960 (0.01)** | 0.929 (0.02) | 0.951 (0.02) |
| Border | **0.985** | 0.976 | 0.94 | 0.947 | 0.884 | 0.88 | 0.375 | 0.944 | 0.955 (0.05) | 0.957 (0.02) | 0.943 (0.03) |
| Overlap | **0.817** | 0.788 | 0.782 | 0.811 | 0.727 | 0.748 | 0.679 | 0.675 | 0.754 (0.01) | 0.734 (0.01) | 0.770 (0.01) |
| Outdoor | 0.864 | 0.823 | 0.342 | 0.826 | 0.685 | 0.733 | 0.18 | 0.65 | 0.958 (0.03) | **0.963 (0.02)** | 0.942 (0.02) |
| COIL | 0.754 | 0.663 | 0.666 | 0.791 | 0.587 | 0.631 | 0.096 | 0.702 | **0.851 (0.03)** | 0.859 (0.01) | 0.832 (0.02) |

Table 2: Averaged online accuracy of RRNADE over 5 seeds with standard deviations, compared with (Losing et al., 2018)

| Accuracy | L++.NSE | DACC | LVGB | KNNs | KNNwa | SAM | SPOK | RRNADE -2RNN | RRNADE -LSTM | RRNADE -GRU |
|---|---|---|---|---|---|---|---|---|---|---|
| CoverType | 0.850 | 0.899 | 0.909 | 0.958 | 0.932 | 0.952 | 0.883 | **0.960 (0.01)** | 0.929 (0.02) | 0.951 (0.02) |
| Electricity | 0.728 | 0.831 | 0.832 | 0.713 | 0.739 | 0.825 | 0.742 | 0.864 (0.02) | **0.878 (0.01)** | 0.858 (0.01) |
| Outdoor | 0.422 | 0.644 | 0.601 | 0.86 | 0.837 | 0.888 | 0.810 | 0.958 (0.03) | **0.963 (0.02)** | 0.942 (0.02) |
| Poker Hand | 0.779 | 0.790 | **0.864** | 0.829 | 0.721 | 0.816 | 0.731 | 0.847 (0.02) | 0.851 (0.02) | 0.840 (0.01) |
| Rialto | 0.596 | 0.711 | 0.604 | 0.772 | 0.750 | 0.814 | 0.618 | 0.887 (0.05) | **0.936 (0.04)** | 0.915 (0.04) |
| Weather | 0.771 | 0.732 | 0.781 | 0.785 | 0.769 | 0.783 | 0.741 | 0.786 (0.00) | **0.790 (0.01)** | 0.783 (0.01) |

Table 3: Averaged online accuracy of RRNADE over 5 seeds with standard deviations, compared with (Coelho & Barreto, 2022)

## 4.2 CLASSIFICATION

In this subsection, we present experiment results for online classification on various classic benchmarks of online classification and compare RRNADE to both density based as well as non density based methods. The results of these methods are obtained from the corresponding papers. To keep the fairness, we conduct data preprocessing and evaluation procedure the same way as mentioned in each of these papers. We validate on the first 1,000 examples or first $10\%$ of the data (whichever is smaller) to select the number of mixture components, number of hidden states, window size $l$ as well as likelihood length $p$, while $l$ is set to be no larger than 10. The input encoder is still a one layer fully connected neural networks with number of neurons being one of the hyperparameters as well. This validation routine is consistent with all three papers we compare with.

First, we compare with several density based classifiers listed in (Chen et al., 2021). Note the "Cover type", "Poker hand", "Transient chessboard", "Rialto" as well as "Mixed drift" are originally multi-class data streams. Following the same procedure as in (Chen et al., 2021), we generate their binary versions by extracting the largest two classes from each data stream, respectively. The AUC scores on various datasets are presented in Table 1. From this table, we can see all RRNADE variants consistently outperforming the other methods. In addition, in many datasets, e.g. "Electricity", "Hyperplane" etc., we outperform the best compared method by a significant margin.

Second, we compare with multiple non-density based classification methods in both (Coelho & Barreto, 2022; Losing et al., 2018). The running average of the classification accuracy is reported in Table 3 and Table 2, respectively. In Table 2, N/A indicates that the corresponding result is not reported in the original paper. We can see that in almost every datasets, we achieve competitive results, if not significantly better. For synthetic datasets, in "Inter RBF", various Gaussians are replacing each other every 3000 samples, representing an abrupt concept drift, while the dataset "Moving RBF" is constructed such that Gaussian distributions with random initial positions, weights and standard deviations are moved with constant speed, representing a gradual concept drift. Here we can see, on both of these datasets, we outperform other methods, further showcasing RRNADE's ability to adapt to concept drift. For real world data, "Weather", "Electricity", "Outdoor", "COIL" as well as "Rialto" are all data streams with sequential dependencies. Here we can also see that on these datasets, RRNADE outperforms other compared methods.

## 4.3 ABLATION STUDY

In this ablation study, we are looking into the significance of RRNADE's two components, i.e. the density module $\xi$ and the recurrent module $g$. We compare RRNADE with three baselines: RRNADE without recurrent module (NR), RRNADE without density module (ND), as well as RRNADE without both recurrent and density module (NRND). For NR, we replace the recurrent

| Accuracy | RRNADE-2RNN | RRNADE-LSTM | RRNADE-GRU | NR | NRND | ND-LSTM | ND-2RNN | ND-GRU | |
|---|---|---|---|---|---|---|---|---|---|
| Border | 0.96 (0.05) | **0.96 (0.02)** | 0.94 (0.03) | 0.90 (0.03) | 0.80 (0.02) | 0.81 (0.02) | 0.80 (0.02) | 0.81 (0.01) |  |
| Overlap | 0.75 (0.01) | 0.73 (0.01) | **0.77 (0.01)** | 0.75 (0.03) | 0.65 (0.02) | 0.64 (0.01) | 0.65 (0.02) | 0.66 (0.02) | |
| Inter RBF | 0.90 (0.02) | 0.92 (0.02) | **0.92 (0.03)** | 0.87 (0.02) | 0.46 (0.05) | 0.87 (0.09) | 0.66 (0.08) | 0.79 (0.04) | |
| Moving RBF | 0.74 (0.01) | 0.74 (0.02) | **0.77 (0.01)** | 0.51 (0.02) | 0.42 (0.01) | 0.59 (0.03) | 0.44 (0.02) | 0.58 (0.04) | |
| Outdoor | 0.96 (0.03) | **0.96 (0.02)** | 0.94 (0.02) | 0.95 (0.01) | 0.31 (0.02) | 0.42 (0.02) | 0.45 (0.03) | 0.39 (0.02) | |
| Weather | 0.79 (0.00) | **0.79 (0.01)** | 0.78 (0.01) | 0.78 (0.01) | 0.78 (0.01) | 0.79 (0.01) | 0.78 (0.01) | 0.79 (0.0) | |
| Rialto | 0.89 (0.05) | **0.94 (0.04)** | 0.92 (0.04) | 0.82 (0.04) | 0.73 (0.01) | 0.92 (0.04) | 0.83 (0.05) | 0.93 (0.03) | |

Figure 3: Comparison of RRNADE with three baselines NR, ND, NRND

module of RRNADE by a two layered fully connected neural networks, which only takes the data at the current and previous time step as the input. For ND, we replace the density module of RRNADE by a two layered fully connected neural networks, while for NRND, a two layered neural network is used to map from the current step input to its label. To correspond to the three variants of RRNADE, ND also has three different recurrent units, namely GRU, LSTM, and 2RNN. A graphical illustration of the baseline models can be seen in Figure 4 in the Appendix. The hyperparameters are selected using validation in the same fashion as we mentioned before. We present the results in Table 3.

From the table, we can see that on all examined datasets, RRNADE outperforms all baselines. In some datasets, e.g. "Moving RBF" and "Inter RBF", only by using both of the modules, i.e. the density module $\xi$ and the recurrent module $g$, can the model reach to the best performance. However, on some other datasets, e.g. "Rialto", the missing of the recurrent module alone is detrimental, while for datasets like "Overlap", the density module is of great importance. On one hand, recurrent module captures the temporal relations in the data, which in some cases are the key to predicting the correct label. In the top right figure of Figure 3, we show the learning curves of the three variants of RRNADE and NR on "Rialto" dataset. We can see clearly that NR converges slower and to a worse solution compared to RRNADE. This is due to the fact that the label has a specific ordering, which, if not using the recurrent module, is hard to capture. On the other hand, the use of density module provides a proper inductive bias in some cases, which accelerate the convergence and improve the final solution. For example, both "Inter RBF" and "Moving RBF", as explained earlier, are generated from Gaussians with concept drifts. Therefore we can see that they overall have much better performance with the density module. In addition, we train the NRND model on "overlap" dataset in an offline fashion using gradient descent with batch size 1 without random permutation of the data. Note under this training routine, the first epoch is equivalent to learning under streaming setting. In bottom right figure of Figure 3, we show the learning curves of such trained NRND model of the first 10 epochs (the increasing color gradient corresponds to the increase in the number of epochs) as well as of NR and RRNADE-GRU. As the number of epochs increases, NRND slowly reaches to the performance of NR. However, in the first epoch, i.e. the online setting, we can see although all the methods start at close positions, the ones with density module improve much faster.

## 5 CONCLUSION

In this paper, we propose the Recurrent Real-valued Neural Autoregressive Density Estimator (RRNADE), an extension of the classic RNADE model to its online setting. The core idea of RRNADE is to maintain a set of sufficient statistics for the future via recurrent function (recurrent module) and approximate the conditional density function using mixture of Gaussian (density module), parameterized by neural networks. We show that theoretically, RRNADE is strictly more expressive than the Gaussian hidden Markov model, which is a classic model for learning sequential data. We then propose learning algorithms for using RRNADE on online density estimation and classification of data streams, respectively. For the empirical studies, we conduct experiments on synthetic data showing RRNADE is able to learn the density function parameterized by a Gaussian HMM and RRNADE is efficient in adapting to concept drifts. For the classification tasks, we compare RRNADE with various methods on multiple synthetic and real world datasets. We show that RRNADE outperforms all the methods on almost every dataset. In the ablation study, we further showcase the importance of both the recurrent module and the density module, where the recurrent module helps capture the sequential dependencies of the data stream, while the density module helps with the online optimization. For the future work, recall that we use uniform distribution as the prior for RRNADE on classification tasks. In the future, we would like to investigate a more adaptive way of estimating the prior. In addition, since our model is a density model, it would be interesting to investigate the possibility of online clustering.

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

# A  APPENDIX

## A.1  PROOF OF THEOREM

**Theorem A.1.** *Given a Gaussian HMM with $k$ states $\eta = \langle \boldsymbol{\mu}, \mathbf{T}, O \rangle$, where $O : \mathbb{R}^k \times \mathbb{R}^d \to \mathbb{R}_0^+$ is the Gaussian emission function, $\boldsymbol{\mu} \in \mathbb{R}^k$ is the initial state distribution and $\mathbf{T} \in [0,1]^k$ is the transition matrix, there exists a $k$ states $k$ components RRNADE $\mathcal{R}$ with full covariance matrices such that the density function over all possible trajectories sampled by $\eta$ can be computed by $\mathcal{R}$: $p^\eta(\boldsymbol{x}_1 \cdots \boldsymbol{x}_n) = \prod_{i=1}^n f_\mathcal{R}(\boldsymbol{x}_{\leq n})$ for any trajectory $\boldsymbol{x}_1 \cdots \boldsymbol{x}_n$. Moreover, there exists an RRNADE $\mathcal{R}'$ such that no Gaussian HMM can compute its density.*

*Proof.* For the Gaussian HMM $\eta$, given an observation sequences $\boldsymbol{x}_1 \cdots \boldsymbol{x}_n$, its density under $\eta$ is:

$$p^\eta(\boldsymbol{x}_1 \cdots \boldsymbol{x}_n) = O(\boldsymbol{m}^\top, \boldsymbol{x}_1) O(\boldsymbol{m}^\top \mathbf{T}, \boldsymbol{x}_2) \cdots O(\boldsymbol{m}^\top \mathbf{T}^{n-1}, \boldsymbol{x}_n),$$

where $O(\boldsymbol{h}, \boldsymbol{x}) = \sum_{i=1}^k \boldsymbol{h}_i \mathcal{N}(\boldsymbol{x}|\boldsymbol{\mu}, \boldsymbol{\Sigma})$ for some mean vector $\boldsymbol{\mu}$ and covariance matrix $\boldsymbol{\Sigma}$. Let $\mathcal{R} = \langle \boldsymbol{h}_0, g, \xi, \phi \rangle$ be an RRNADE, more specifically, let $\boldsymbol{h}_0 = \boldsymbol{m}$, $g(\boldsymbol{h}, \boldsymbol{x}) = \mathcal{A} \bullet_1 \boldsymbol{h}^\top \bullet_2 \phi(\boldsymbol{x})$ $\mathcal{A}_{:,i,:} = \mathbf{T}$ for $i \in [k]$, $\phi(\boldsymbol{x}) = [\frac{1}{k}, \frac{1}{k}, \cdots, \frac{1}{k}]^\top$ and $\xi = O$. Note it reasonable to let $\xi = O$, since as long as we let $\boldsymbol{\beta}_i = \boldsymbol{h}_0^\top \mathbf{T}^{i-1}$, $\boldsymbol{\beta}_0 = \boldsymbol{h}_0^\top$, $\boldsymbol{\mu}_i = \boldsymbol{\mu}$ and $\boldsymbol{\Sigma}_i = \boldsymbol{\Sigma}$, following equations 4, then for any $\boldsymbol{h} \in \mathbb{R}^k, \boldsymbol{x} \in \mathbb{R}^d$, we have $\xi(\boldsymbol{h}, \boldsymbol{x}) = O(\boldsymbol{h}, \boldsymbol{x})$. Then under this parameterization, we have $\mathcal{A} \bullet_2 \phi(\boldsymbol{x}_j) = \mathbf{T}$. Then the RRNADE computes the following function:

$$f_\mathcal{R}(\boldsymbol{x}_1, \cdots, \boldsymbol{x}_i) = \xi((\mathcal{A} \bullet_1 \boldsymbol{h}_0^\top \bullet_2 \phi(\boldsymbol{x}_1))^\top (\mathcal{A} \bullet_2 \phi(\boldsymbol{x}_2)) \cdots (\mathcal{A} \bullet_2 \phi(\boldsymbol{x}_{i-1})), \boldsymbol{x}_i)$$
$$= \xi(\boldsymbol{h}_0^\top \mathbf{T}^{i-1}, \boldsymbol{x}_i) = O(\boldsymbol{m}^\top \mathbf{T}^{i-1}, \boldsymbol{x}_i)$$

Therefore, we have:

$$p^\eta(\boldsymbol{x}_1 \cdots \boldsymbol{x}_n) = O(\boldsymbol{m}^\top, \boldsymbol{x}_1) O(\boldsymbol{m}^\top \mathbf{T}, \boldsymbol{x}_2) \cdots O(\boldsymbol{m}^\top \mathbf{T}^{n-1}, \boldsymbol{x}_n)$$
$$= f_\mathcal{R}(\boldsymbol{x}_1) f_\mathcal{R}(\boldsymbol{x}_1, \boldsymbol{x}_2) \cdots f_\mathcal{R}(\boldsymbol{x}_1, \cdots, \boldsymbol{x}_n) = \prod_{i=1}^n f_\mathcal{R}(\boldsymbol{x}_{\leq n})$$

For the proof of the second half of the theorem, consider a shifting Gaussian HMM, where the mean vector of the Gaussian emission is a function of the time steps, i.e., $\boldsymbol{\mu} = q(i)$, where $i = 1, 2, \cdots$. For simplicity, assume the shifting Gaussian HMM is for one dimensional sequences and has one components. In addition, let $q(i) = i$ and assume the variance is 1. Then the emission function can be written as $O^t(x) = \mathcal{N}(x|t, 1)$. Then the density of a sequence $x_1, \cdots, x_n$ under this shifting Gaussian HMM $\eta_s$ is:

$$p^{\eta_s}(x_1, \cdots, x_n) = O^1(x_1) O^2(x_2) \cdots O^n(x_n).$$

We show that this density cannot be computed by a Gaussian HMM of finite states. If $p^{\eta_s}$ can be computed by a Gaussian HMM, then for the mean vector $\boldsymbol{\mu}$ there exists an initial weight vector $\boldsymbol{m}$, a transition matrix $\mathbf{T}$ satisfying the following linear system:

$$\begin{cases} \boldsymbol{m}^\top \boldsymbol{\mu} & = 1 \\ \boldsymbol{m}^\top \mathbf{T} \boldsymbol{\mu} & = 2 \\ \quad \vdots \\ \boldsymbol{m}^\top \mathbf{T}^{n-1} \boldsymbol{\mu} & = n \\ \quad \vdots \end{cases}$$

This linear system is, however, overdetermined, as $\boldsymbol{\mu}$ is a vector of finite size, while there are infinite linearly independent equations to satisfy. Therefore, a Gaussian HMM of finite states cannot compute the density function of a shifting Gaussian HMM.

We now show such density can be computed by a RRNADE. Let $\boldsymbol{h}_0^\top = [1, 1]$, and $\mathcal{A}_{:,i,:} = \begin{bmatrix} 1 & 1 \\ 0 & 1 \end{bmatrix}$, $\mathbf{M}_i = \langle \boldsymbol{h}_{i-1}, [0, 1] \rangle$, $\phi(x)^\top = [0.5, 0.5]$, $\boldsymbol{\Sigma}_i = 1$. Then we have:

$$f_\mathcal{R}(x_1, \cdots, x_i) = \xi((\mathcal{A} \bullet_1 \boldsymbol{h}_0^\top \bullet_2 \phi(x_1))^\top (\mathcal{A} \bullet_2 \phi(x_2)) \cdots (\mathcal{A} \bullet_2 \phi(x_{i-1})), x_i)$$
$$= \xi(\boldsymbol{h}_0^\top \mathbf{T}^{i-1}, x_i) = \xi([1, i], x_i) = \mathcal{N}(x_i|i, 1)$$

Therefore:

$$p^{\eta_s}(x_1, \cdots, x_n) = \mathcal{N}(x_1|1,1)\mathcal{N}(x_2|2,1)\cdots\mathcal{N}(x_n|n,1)$$
$$= f_{\mathcal{R}}(x_1)f_{\mathcal{R}}(x_1,x_2)\cdots f_{\mathcal{R}}(x_1,\cdots,x_n) = \prod_{i=1}^{n} f_{\mathcal{R}}(\boldsymbol{x}_{\leq n})$$

Therefore, for the given shifting Gaussian HMM density, it can be computed by a RRNADE, but cannot be computed by a Gaussian HMM with finite states. □

### A.2 THREE BASELINE MODELS FOR ABLATION STUDY

For NR, we replace the recurrent module of RRNADE by a two layered fully connected neural networks, which only takes the data at the current and previous time step as the input. For ND, we replace the density module of RRNADE by a two layered fully connected neural networks.

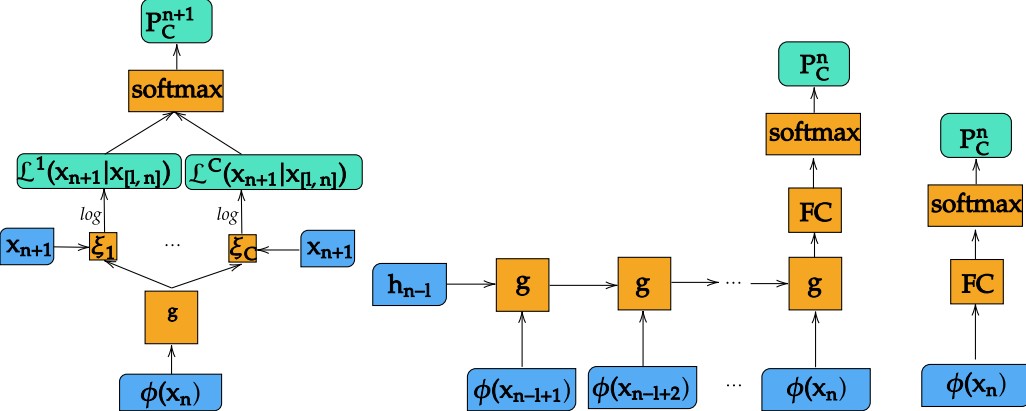

Figure 4: The three baseline models NR (left), ND (middle), NRND (right).

