# OpenReview forum: "Recurrent Real-valued Neural Autoregressive Density Estimator for Online Density Estimation and Classification of Streaming Data"
_ICLR.cc/2023/Conference — Submitted to ICLR 2023_

### Official Review · Reviewer_qnzA · 2022-10-24

**Confidence:** 3
**Correctness:** 2
**Technical Novelty And Significance:** 2
**Empirical Novelty And Significance:** 1
**Recommendation:** 3

**Clarity, Quality, Novelty And Reproducibility:**

The paper is easy to follow.  The overall quality and clarity are generally below the bar of ICLR publication. Also, the novelty is limited since the used techniques are widely-used things while the authors do not show advanced extension or new philosophy. As for the experimental study, more comparison is needed, although I trust that the results can be reproduced.

**Strength And Weaknesses:**

Strengths:
+ The motivation to use a recurrent module and a mixture of Gaussian (parameterized by NNs) is interesting;
+ The theoretical result verifies, in theory, the potential merits of RRNADE

Weaknesses:
+ The technical contribution is generally limited since RNADE and recurrent models (RNN as a representative) are existing works. Authors do not show significant novelty in framework/algorithmic development
+ Theorem 3.1 is not easy to understand. What is the assumption behind the theory? After presenting mathematically the theorem, further remarks and/or discussion on its meaning and usefulness are needed. This may help readers follow the key point behind the theoretical result.
+ It is not clear to what extent and how the proposed RRNADE outperform RNADE. More discussion and empirical studies on clarifying this issue should be beneficial to enhance the contribution.

**Summary Of The Paper:**

This paper studies the problem of online density estimation and classification of streaming data. A new model named RRNADE (Recurrent Real-valued Neural Autoregressive Density Estimator) is proposed for problem-solving. In RRNADE , a recurrent module is used to maintain a set of sufficient statistics for the future state, and a neural net-based mixture of Gaussian module is used to approximate the conditional density function. The authors also prove theoretically that RRNADE is strictly more expressive than the Gaussian hidden Markov model. Experimental studies on both synthetic and real-world datasets are conducted to show the effectiveness of RRNADE.

**Summary Of The Review:**

The paper does tackle an important problem for stream data modeling. However, the technical contribution is not strong enough to support acceptance. So it is unclear to me whether the contribution of this paper is significant. The paper should also consider more datasets and baselines. Thus, overall I can not recommend acceptance.

---

### Official Review · Reviewer_BvnL · 2022-10-27

**Confidence:** 4
**Correctness:** 4
**Technical Novelty And Significance:** 2
**Empirical Novelty And Significance:** Not applicable
**Recommendation:** 3

**Clarity, Quality, Novelty And Reproducibility:**

## Quality
The paper is not well written. Typos need to be carefully corrected.

## Novelty
As mentioned, I do not think combining two existing techniques is novel enough.

## Reproducibility
Necessary description are mentioned.


**Details Of Ethics Concerns:**

Not applicable.

**Strength And Weaknesses:**

## Strength
The paper extends a previous work, RNADE, to the online learning setting. The algorithm is clearly stated and easy to understand. The experiment includes many baseline algorithms to make a detailed comparison.

## Weakness
The novelty of the proposed algorithm is limited. The proposed algorithm is very similar to an existing work, RNADE. Specifically, based on Eqn. (1)~(5), the main difference between RRNADE and RNADE is the usage of the recurrent unit. Additionally, RNN is a popular tool in the online learning literature. I don't think combining the two technologies is innovative enough to meet ICLR's standards.

The experiment results are somewhat tricky. Specifically, the performance is highly related to the RNN type, which leads to some tricky results. For example, in line "Cover type" of Table 1, the usage of "2RNN" seems to be necessary, while "GRU" appears to be necessary sometimes (line "Moving RBF" in Table 2). Doing so is unfair to other baseline algorithms because their structures are fixed rather than carefully tuned by the authors. Meanwhile, the details of how the baseline algorithms are implemented should be written in detail. Another question is how we choose the type of RNN in reality.

The writing needs to be polished. After reading the first paragraph in the introduction, I feel like I am reading an early-stage draft. For example, the "he" should be "The", and the punctuation is missing in the last sentence. The current version is apparently not ready to be published on ICLR.

**Summary Of The Paper:**

This paper considers the problem of density estimation and streaming data classification in the online learning setting. It proposes a new algorithm -- Recurrent Real-valued Neural Autoregressive Density Estimator (RRNADE), which combines recurrent module and existing work (RNADE). The paper provides some theoretical results and detailed experiment results to show the effectiveness of the proposed algorithm.

**Summary Of The Review:**

Considering the lack of innovation, the tricky experimental results, and some typos, I tend to give a rejection.

---

### Official Review · Reviewer_jzmZ · 2022-10-28

**Confidence:** 4
**Clarity, Quality, Novelty And Reproducibility:** The proposed approach and results pre…
**Correctness:** 3
**Technical Novelty And Significance:** 2
**Empirical Novelty And Significance:** 1
**Recommendation:** 3

**Strength And Weaknesses:**

Strengths:
- Paper describes the proposed approach well conceptually. The detailed update equations clearly illustrates the contribution.
- Empirical results show good performance of the proposed method on benchmark stream datasets typically used in data stream papers.

Weaknesses:
- The proposed approach seem simple compared to the existing RNADE method. It would be better to discuss problems with data stream when using RNADE with a bit more deeper insight rather than stating the 2 reasons provided. Moreover, the reason of number of parameter size going to infinity in RNADE seems like a stretch.
- Empirical evaluation is not strong enough to show efficacy of the proposed approach. It would be better if more complex datasets are used apart from the benchmark datasets, particularly using image or text datasets as mentioned in the paper's motivation. Furthermore, the competitive approaches used for classification does not seem to be systematically chosen. For example, in the classification scenario, there are multiple non-density based data stream classifiers showing better results on the same dataset as reported in Table 2 and 3 such as Lv, Yanxia, et al. "A classifier using online bagging ensemble method for big data stream learning." Tsinghua science and technology 24.4 (2019): 379-388. It would be better to compare with better approaches than just the 2 papers cited in the table.

**Summary Of The Paper:**

The paper describes a proposal to enhance real-valued neural autoregressive density estimator (RNADE) by utilizing RNN-based mechanisms to be applicable over a concept-drifting data stream for density estimation and classification tasks. Using RNNs for estimating mean and variance, and constraining the number of parameters, the authors tweek the RNADE method for online settings. Empirical results over benchmark stream datasets show that the proposed approach performs best in majority of cases, both for density estimation and classification tasks, even under synthetically generated concept drifts.

**Summary Of The Review:**

Overall, though the proposed approach seemingly works well on typical benchmark datasets, the quality of evaluation needs to dramatically improve, particularly with the use of better datasets (since a NN is being used), and better competitive approach should be used for comparison. Furthermore, the paper is missing key details such as cold-start problem, drift detection etc.

---

> ### Author Response · Authors · 2022-11-15
> **Reply**
>
> We thank the reviewer for their reviews.
>
> 1. The reviewer mentioned that more examination of the RNADE method under the streaming setting would be beneficial. We agree and we will add corresponding experiments in the future revision. The reviewer also mentioned that the reason for the number of parameter sizes going to be infinite in RNADE seems like a stretch. Could the reviewer elaborate on this?
>
> 2. The reviewer mentioned it would be better to use more complex datasets such as image or text datasets. Does the reviewer have some specific datasets in mind that suit online density estimation/classification under the streaming setting?
>
> 3. We thank the reviewer's mentioning of Lv, Yanxia, et al. in Tsinghua science and technology. We were unaware of the paper. However, the method proposed in this paper is not a standalone algorithm but rather an ensemble method, which leverages online bagging of multiple classifiers to make the final prediction. This ensemble scheme can be easily adapted to our methods as well to further improve on the existing results.
>
> 4. For the cold-start problem, in the experiments we use the first 1000 examples to validate the hyperparameters. Drift detection should be Incorporated within the algorithm, i.e. the trade-off of l and p.

---

### Official Review · Reviewer_ybf6 · 2022-10-31

**Confidence:** 3
**Correctness:** 3
**Technical Novelty And Significance:** 3
**Empirical Novelty And Significance:** 3
**Recommendation:** 5

**Clarity, Quality, Novelty And Reproducibility:**

The writing quality of the paper is fine. But there seems to be some important details missing (please see the points I raised in the Weaknesses section above). The paper only gives references for some of the experimental setups. The code is not provided either. These may hurt the reproducibility.
Below are other points unclear to me.
- The role of the window defined by $l$ is not clear. In Algorithm 1, $l$ does not seem to have any effect on the procedures. (Note that $f_R(x_{t-l+1}, \dos, x_t)$ is defined as $\xi(h_{t-1}, x_{t})$ according to Eq. (5).
- In page 5, "the first observation $x_1$ is discarded but $h_1$ still represents sufficient statistics of all past observations, including $x_1$": is this an assumption?
- It was not clear how we model the density without the density module in ND and NRND.

**Strength And Weaknesses:**

# Strengths
- The results for the experiments are impressive.
- The authors point out an important weakness of the existing model, RNADE, on which the proposed model is built: it only allows an additive update to the hidden state. The proposed method improves on this point.
- They prove that the proposed model is strictly more expressive than Gaussian HMMs.

# Weaknesses
- The proposed method has several hyper-parameters that are not systematically selected in the experiments.
- The proposed method introduces an additional complexity the existing model by introducing a flexible recurrent module.
- The main contribution of the work is not clear to me from the paper. It seems RNADE can be almost directly applicable to the online setup. The difference I can see from RNADE is removing the limitation of the recurrent module being additive and allow it to be any function (the first eq. of Eqs. (3)). But the paper does not provide enough evidence showing that the modification is really worth the complication.
- The problem setup is not mathematically defined. It is not clear what distribution drift or what evaluation metric the paper considers. Without defining those, I don't think we can even discuss the learnability issue. They use terms such as "abrupt concept drift" or "smooth drift", but they are not clearly defined.
- The paper does not explain how and how much the update (line (7) and (12) of Algorithm 1) is performed. For example, the learning rate should matter in online learning.

**Summary Of The Paper:**

This paper proposes a method for learning data probability density functions in an online manner. The proposed method introduces a recurrent module into a neural Gaussian mixture density model. The method takes into account temporal dependencies by allowing the output depend on the past observations within a fixed window. It also allows us to tune how adaptive the model is by specifying the window over which the likelihood is maximized. The authors show that the proposed model is strictly more expressive than Gaussian HMMs. The experiments on the classification application show that the proposed method often outperforms existing methods.

**Summary Of The Review:**

The idea of introducing a flexible recurrent module to RNADE seems interesting, and the resulting proposed method is effective according to the experiments. However, the paper is not clear enough and does not present the contribution in a convincing way. I think the paper falls a little below the acceptance bar for those reasons.

---

> ### Author Response · Authors · 2022-11-15
> **Reply**
>
> We thank the reviewer for their reviews.
>
> 1. For the hyper-parameters, we mentioned in the experiment section that these will be selected via validation on the first 1,000 time steps.
>
> 2. The main difference between RNADE and RRNADE is that RNADE does not have a recurrent structure. Therefore under the streaming data setting, the number of the parameters for RNADE will grow indefinitely. We agree that adding more experiments to showcase that RNADE will fail under this setting would be beneficial for the paper.
>
> 3. The reviewer mentioned that in Algorithm 1, the window size does not affect the procedures. We would like to point out that the log likelihood $\mathcal{L}_j^c$ is computed for the past $l$ (window size) steps.
>
> 4. The reviewer was wondering if in page 5, ”the first observation is discarded but still represents sufficient statistics of all past observations, including ” is this an assumption. The answer is no. However, we are a bit confused about this question. Could the reviewer elaborate more on their concerns?
>
> 5. For ND and NRND, the point is to see when stripping the density module away, how would the remaining structure work, for the ablation purpose, and the experiments are done to evaluate the classification task. Therefore, for ND and NRDE, we do not model the density, but rather directly do classification through a softmax layer in the end.

---

> > ### Comment · Reviewer_ybf6 · 2022-11-16
> > **Elaboration on point 4**
> >
> > I thank the authors for the response to my comments.
> >
> > I would like to elaborate on the following point:
> > > The reviewer was wondering if in page 5, ”the first observation is discarded but still represents sufficient statistics of all past observations, including ” is this an assumption. The answer is no. However, we are a bit confused about this question. Could the reviewer elaborate more on their concerns?
> >
> > I meant that the claim "$h_1$ still represents sufficient statistics of all past observations, including $x_1$" is a strong mathematical claim and needs a proof.

---

### Official Review · Reviewer_7rkR · 2022-10-31

**Confidence:** 3
**Correctness:** 3
**Technical Novelty And Significance:** 2
**Empirical Novelty And Significance:** 2
**Recommendation:** 3

**Clarity, Quality, Novelty And Reproducibility:**

The paper is clear and easy to follow. The proposed method combines existing methods RNADE and reccurent component to create a *new* model which is better suited for tasks with streaming data. The paper provides most details to reproduce the experiments. However, I could not find any *direct* reference to the code and the datasets in the paper.

**Strength And Weaknesses:**

Pros:

1. The task of online classification and density estimation is important and well motivated.
2. The method is simple, clear, and brings some practical improvements. The model section is easy to follow.

Cons:

- Related works: I feel that the related work section could be more detailed. The authors mention continual and increamental leanring as related problems but these are not discussed in the related work. It might also be interesting to discuss the related works around other density estimation methods (e.g. Normalizing Flows). Further the concept of drift could also be more detailed. E.g. (Generalized Out-of-Distribution Detection, Yang et al) makes the distinction between sensory/covariate and semantic shifts. Action suggestion: Enrich the related work wth some discussion on continual learning, density estimation methods, and shift types.
- Mathematical notation: I was confused with the not bold x notation in formula (1), (2) vs the bold x notation in defintion 1. What are the difference between these two notations ? I was also confused with the argument of f_R in the definition 1 which looks like a multiplication of all x_1, x_2, … and x_n. Should it be f_R(x_1, x_2, x_n). Action suggestion: Fix and clarify the mathematical notations.
- Experiences:
    - The importance of the hyper-parameters l and p is not clear. It would be interesting to have one experiment where these hyper-parameter are changing. It allows practionners to know what is the impact of these introduced hyper-parameters. Action suggestion: Make an experiment when l and p vary.
    - I had the feeling that the test in Fig. 2 text was very small and not easy to read withou important zooming. Action suggestion: Make the text bigger.
    - The experiments on density estimation focus on only on synthetic toy datasets. While these experiments are interesting since they provide the ground-truth, I feel that it is important to test also on real-datasets. Further, it would also be important to compare to other methods. Currently results only show RRNADE results on Fig. 2. Action suggestion: add real-dataset and other baselines in the density estimation experiments.
    - For completeness, it would be nice to have details on the used datasets in the paper or appendix. Further, these datasets look fairly small datasets and require only very simple input encoder. E.g. RNADE paper look also at image and maybe speech datasets. Action suggestion: consider larger datasets with more complex input encoder.
    - Since RRNADE is an extension of RNADE, it would be interesting to have some comparison of the two methods in the experiments to show that RRNADE fixes practical limits of RNADE (e.g. number of parameters). Action suggestion: Make a comparison of RRNADE and RNADE.
    - “The results of these methods are obtained from the corresponding papers.”: It sounds that the the baselines results are simply copy-pasted from the previous papers. I feel that this is generally a bad practice. For a precised evaluation, methods should be implemented using the same framework and use the same evaluation pipeline. In partciular, it is imposible to know how ISVM and LASVM perform compare to RRNADE on four datasets. Further, the baselines do not have error bars. Action suggestion: Ideally, run each baseline on each datasets and get error bars. If this is too difficult in the available time, I would expect the repoduction of at least the two strongest baselines and complete the N/A missing numbers.

Others:

- Typo: “**T**he problem” first paragraph on p.1
- Typo: “constrain**t**s” second paragraph on p.1
- Cite RNADE paper  when introducing it on p.1

I am happy to improve my score if a majority of the above points are addresses (e.g. with the action suggestions).

**Summary Of The Paper:**

The paper looks at the task of online density estimation and classification. This is motivated by the problem of memory constraints, concept drifts, and temporal correlations. To this end, they propose Recurrent Real-valued Neural Autoregressive Density Estimator (RRNADE) which is an extension of the previous work RNADE with a recurrent neural network component. This new model is theoretically more expressive than Gaussian HMM and empiracally better than selected baselines.

**Summary Of The Review:**

Overall, I vote for reject. The task is well motivated and the model is clearly presented. My major concerns are about clarifications on the related work and the experiences (see cons). Hopefully the authors can address my concern in the rebuttal period.

---

> ### Author Response · Authors · 2022-11-15
> **Reply**
>
> We thank the reviewer for the detailed feedback! Below we will clarify and answer some of the reviewer's concerns.
>
> 1. We thank the reviewer for mentioning other related works. This is a great point and the papers suggested are quite relevant. We will further enrich the related work section in the future revision.
>
> 2. The reviewer mentioned a great point on the importance of the hyper-parameters l and p. Intuitively, l controls the temporal relations of the sequence while p acts as an indicator for (abrupt) concept drift. For example, if the sequence has frequent abrupt concept drift, then p ought to be small to adapt the model to this drift more quickly. In addition, if the sequence has long time dependencies, then l needs to be big to capture it. Therefore maybe another ablation experiment on a toy dataset where we can control the temporal relations as well as introduce various drifts can help illustrate this intuition more clearly. We are also looking into how to adjust p adaptively, as the type of concept drifts can change in the data stream as well, thus might require different p at different time steps.
>
> 3. The reviewer mentioned incorporating real datasets and other methods for the online density estimation experiment parts. Does the reviewer have something specific in mind? We are also looking into adding these but rarely found proper datasets or methods with the corresponding code base.
>
> 4. We will include a description of the datasets. Just to clarify, many of the datasets we examined are large, consisting of $O(10^5)$ number of examples with many features $O(10)$. Similarly to the previous point, does the reviewer have some suggestions on other suitable datasets?
>
> 5. It is also a good point for the comparison of RRNADE and RNADE. We mentioned the impracticality of RNADE under the online setting for streaming data. However, we agree it would make the point clear by incorporating a toy experiment or adding RNADE as one of the baselines in all our experiments.
>
> 6. As for the copy-pasting experiment results from previous papers, we do agree that implementing these methods and running them again to obtain the error bars are useful. However, for most of the compared methods, we couldn't find proper implementations. In addition, in most of the experiments, RRNADE outperforms other compared methods by a large margin.

---

> > ### Comment · Reviewer_7rkR · 2022-11-16
> > **Thank you for your answer**
> >
> > Thank you for your answer. I feel that the ideas mentioned by the authors are good direction to improve the paper. Here would be some further thoughts/suggestions to improve the paper on points 3./4. and 6.:
> >
> > 3./4. I would suggest ot provide the details on the used datasets somewhere in the paper or appendix. Another suggestion would be to create semi-synthetic dataset to transform standard real dataset (e.g. CIFAR) into an streaming task.
> >
> > 6. I would suggest to explicitly point out which methods do not provide code which make difficult to reproduce results. For these methods, it might unfortunately be necessary to code them or emphasize that these are copy-pasted numbers making them less reliable.

---

### Official Review · Reviewer_fRiQ · 2022-11-02

**Confidence:** 4
**Correctness:** 4
**Technical Novelty And Significance:** 3
**Empirical Novelty And Significance:** 3
**Recommendation:** 5

**Clarity, Quality, Novelty And Reproducibility:**

- As mentioned in the weakness, the paper lacks details about baselines and datasets. Please try to make the work self-contained and provide brief descriptions, even if in the appendix.
- Some reproducibility details are missing: e.g., the learning rates used to train the models or the hyperparameter configurations with which results are reported (obtained after validation).
- The paper can be made clear at various points:
	- Features in NADE vs. Samples in RRNADE: It was not clear to me until sec 3.1, where the problem about infinite length stream is mentioned, that here we are modeling samples in an autoregressive fashion instead of features (as done in the original NADE work). It could be clarified earlier.
	 - From the text, it seems only \xeta_c and R are updated when encountering a sample with label C. However that is not the case (Alg 1, line 12). The authors could use cross-entropy loss due to the uniform prior assumption, and that makes sense too, but these should be clarified in the text.
	 - Fig 1., some subscripts are wrong. E.g.,  p(x_{1+3}|x_{[l,l+2]}) -> p(x_{n-1+3}|x_{[n-l,n-l+2]})


Minor:
- Fig 2: Please use consistent markers between the bottom and top rows.
- Fig 3: Please use the appropriate size for the table and figures.
- There are typos and grammatical errors. Proofread and correct these. For example,
	- Intro (para 1, line 5) he -> .The
	- Related Work (para 1, line 3) -> relies -> rely
	- Background (line ) "we will background"??
	- Para about **Online density estimation and class...** (line 6): classes -> class
	- next para, line 4: changes -> change
	- Sec 4.1 para 2, time steps 210 -> time step 210

**Strength And Weaknesses:**

Strengths:
- This paper addresses an important problem of learning from streaming data, and the proposed method is shown to outperform several baselines.
- The method is easier to understand and simple to follow. However, some parts are unclear, and the writing could be much better (see weakness)

Weaknesses:
- A description of baselines needs to be included; only pointers to previous papers that use these baselines are provided. Authors should make the papers as self-contained as possible.
- Another problem is that most baselines use non-deep learning methods. Authors do not compare/cite deep learning methods for online learning, such as [1,2]. How does your method compare to these?
- The complexity of tasks is unclear. Mainly authors have not provided any description (e.g., # features, # samples, what is being predicted, etc.) of the datasets, which makes it harder to judge the complexity of tasks.
- Theoretically, RRNADE can solve multi-class classification problems. However, it is only empirically evaluated on binary classification tasks. Would we see similar empirical improvements on multi-class problems?


[1] Online Deep Learning: Learning Deep Neural Networks on the Fly, Sahoo, et al., IJCAI 2018 https://openreview.net/forum?id=SyWKcXMOWH
[2] Adaptive online incremental learning for evolving data streams, Zhang, et al., Applied Soft Computing 105 (2021) https://arxiv.org/abs/2201.01633



**Summary Of The Paper:**

This paper addresses the problem of density estimation and classification on streaming data. The key component of the proposed method is RRNADE, a recurrent extension of NADE. NADE models the density of each feature given previous features, i.e., $p(x^i|x^{<i})$. Similarly, the authors model the density of the current sample given previous samples, i.e., p(x_t|x_{<t}) for the streaming data. They introduce RNN to capture statistics of earlier samples allowing for variable length history. This method maintains a memory of a few past samples. To account for the drift in data distribution, the model is updated with the likelihood of recent samples rather than the entire history that may be available in memory.

RRNADE is extended to predict class labels by learning class conditional density models, and prediction is made with the Bayes rule.

The authors demonstrate the superiority of their method theoretically by showing that RRNADE is more powerful than gaussian HMM and empirically by evaluating on a synthetic dataset for density estimation and several real-world binary classification tasks.


**Summary Of The Review:**

Overall, the proposed method is interesting and relevant. However, many experimental details are missing. Moreover, some important neural network-based baselines are missing. Therefore, I am leaning towards rejecting a paper and would suggest revising the paper. I am willing to increase the score if the authors address the missing baselines and update the manuscript with experimental details.

---

> ### Author Response · Authors · 2022-11-15
> **Reply**
>
> Thank you so much for the detailed reviews. We will take this chance to make a few clarifications on the paper.
>
> 1. The reviewer mentioned that RRNADE is only evaluated on binary classification tasks. However, it is only the case when we compare the results shown in Chen et. al. on density based classification methods. For the other experiments, namely Tables 2 and 3, many of the tasks are multi-class. I think this confusion is caused by the fact that we did not include the description of the datasets, as pointed out by many reviewers. We will include the description of the datasets in future revisions.
>
> 2. The learning rates are determined via validation, the same procedure we described in the paper about other hyperparameters tuning. We will clarify this in the revision.
>
> 3. We thank the reviewer for pointing out the confusion on the features in NADE vs. samples in RRNADE. This is a good point. We will put more emphasis on clarifying this at the beginning.
>
> 4. The reviewer also mentioned that from the text, only $\eta_c$ and $R$ are updated when encountering a sample with label C. We are a bit confused about where this impression comes from. Could the reviewer elaborate a bit more so we can fix this in the future revision?

---

> > ### Comment · Reviewer_fRiQ · 2022-11-22
> > **Elaborating point 2 and 4**
> >
> > Thank you for acknowledging my comments.
> >
> > I want to elaborate further on points 2 and 4 from your comment.
> >
> > > The learning rates are determined via validation, the same procedure we described in the paper about other hyperparameters tuning. We will clarify this in the revision.
> >
> > So this is already clear from the text that hyperparameters were selected by validation over the first few points. However, it is not clear what was the search space for hyperparameters. Further, what were the optimal hyperparameters for each dataset? Both the search space and optimal hyperparameters should be reported.
> >
> > >The reviewer also mentioned that from the text, only $\xi_c$  and $R$ are updated when encountering a sample with label C. We are a bit confused about where this impression comes from. Could the reviewer elaborate a bit more so we can fix this in the future revision?
> >
> > This confusion arises due to the description in section 3. Consider Eq 5 and the last eq on page 5:
> >
> > Eq 5:  $f_R(x_1, \ldots , x_n) = \xi(h_{n−1}, x_n)$
> >
> > Eq on page 5: $p(y_t|x_1 \ldots x_t) \propto p(y_t)p(x_t|x_{<t}, y_t) \simeq p(y_t)f_R(x_{[t−l,t]})$ (The last term in RHS should depend on y, but the dependence is not mentioned --- a minor problem)
> >
> > The text on page 6 elaborates that $f_R$ in above Eq. is $f_{R^c}$, R is shared, and $\xi_c$ are per-class. In any case, there is no mention of normalization in the text. So it appears $f_{R^c}$ is the normalized probability, which caused the confusion that only $R$ and $\xi_c$ will be updated.

---

### Author Response · Authors · 2022-11-15
**Reply to all reviewers**

We thank the reviewers for their detailed reviews. As the reviewers unanimously voted for the rejection of the paper, we do not intend to convince the reviewers to change their scores. However, we would like to take the chance to clarify some concerns raised by the reviewers and discuss potential improvements to the paper.

---

### Decision · Program_Chairs · 2023-01-20

**Decision:**

Reject

**Justification For Why Not Higher Score:**

All reviewers agree that the paper does not meet the standard for publication.

**Justification For Why Not Lower Score:**

N/A

**Metareview: Summary, Strengths And Weaknesses:**

The paper proposes a method for density estimation and classification on streaming data. The method introduces a recurrent module to a neural autoregressive density estimator to model the density of the current sample given the previous samples. It shows improvements over some baselines.
As suggested by the reviewers, the paper will benefit from clarification to provide a more self-contained description of the problem setup and experimental details.